# Combining vertebrate mitochondrial 12S rRNA gene sequencing and shotgun metagenomic sequencing to investigate the diet of the leopard cat (*Prionailurus bengalensis*) in Korea

**Cheolwoon Woo**[1], **Priyanka Kumari**[1,2], **Kyung Yeon Eo**[3], **Woo-Shin Lee**[4], **Junpei Kimura**[5], **Naomichi Yamamoto**[1,2]*

1 Department of Environmental Health Sciences, Graduate School of Public Health, Seoul National University, Seoul, Republic of Korea, 2 Institute of Health and Environment, Graduate School of Public Health, Seoul National University, Seoul, Republic of Korea, 3 Department of Animal Health and Welfare, College of Healthcare and Biotechnology, Semyung University, Jecheon, Republic of Korea, 4 Department of Forest Sciences, College of Agriculture and Life Science, Seoul National University, Seoul, Republic of Korea, 5 College of Veterinary Medicine, Seoul National University, Seoul, Republic of Korea

* nyamamoto@snu.ac.kr

## Abstract

The leopard cat (*Prionailurus bengalensis*), an endangered species in South Korea, is a small feline widely distributed in Asia. Here, we investigated the diet of leopard cats in the inland areas of Korea by examining their fecal contents using vertebrate mitochondrial 12S rRNA gene sequencing and shotgun metagenomic sequencing. Shotgun metagenomic sequencing revealed that the feces were rich in DNA not only of vertebrates but also of arthropods and plants, but care should be taken when using shotgun metagenomic sequencing to identify vertebrates at low taxonomic levels (e.g., genus level), as it was often erroneous. Meanwhile, vertebrate mitochondrial 12S rRNA gene sequencing was found to be accurate in the genus-level identification, as the genera identified were consistent with the Korean fauna. We found that small mammals such as murids were their main prey. By using these two sequencing methods in combination, this study demonstrated that accurate information about the overall dietary content and vertebrate prey of leopard cats could be obtained. We expect that the continued community efforts to expand the genome database of wildlife, including vertebrates, will alleviate the problem of erroneous identification of prey at low taxonomic levels by shotgun metagenomic sequencing in the near future.

## 1. Introduction

The leopard cat (*Prionailurus bengalensis*) is a small feline weighing 1.7–7.1 kg [1]. Due to its adaptability to deforestation and habitat changes, it is distributed in various regions in Asia, such as tropical rainforests, temperate broadleaf forests, coniferous forests, shrub forests, and

**Data Availability Statement:** Raw data of vertebrate mitochondrial 12S rRNA gene sequences is available at NCBI under the BioProject number PRJNA861397. Raw data of shotgun metagenomic sequencing has been published in our previous study [35] under the BioProject accession number PRJNA772888.

**Funding:** This work was supported by the Midcareer Researcher Program of the National Research Foundation of Korea (2020R1A2C1004903) (NY) (https://www.nrf.re.kr/). The funder had no role in study design, data collection and analysis, decision to publish, or preparation of the manuscript.

**Competing interests:** The authors have declared that no competing interests exist.

grasslands [2]. However, in South Korea, its population has declined due to habitat loss, illegal hunting, and road killing, so the Korean Ministry of Environment classified it as a class II endangered species [3, 4]. In addition, it has been pointed out that in South Korea, the risk of regional extinction is high due to their reduced genetic diversity [5]. The leopard cat is the only surviving wild feline species in South Korea and is considered one of the top predators [6]. Therefore, further research on their ecological roles such as food habits is needed to preserve them and ensure their genetic diversity.

The leopard cat is considered to be a hypercarnivore that mainly feeds on rodents, especially murids [7–17]. It was also reported that leopard cats feed on other mammals such as ungulates, lagomorphs, and shrews, as well as birds, reptiles, amphibians, insects, and fishes [7–18]. There is also some literature on the diet of relative species of the leopard cat. For example, the flat-headed cat (*Prionailurus planiceps*) is known to eat fish, but also shrimp, birds, small rodents, and domestic poultry [19]. The fishing cat (*Prionailurus viverrinus*) is a species regarded as a generalist and known to feed on a variety of prey including rodents, birds, and fish [20]. The rusty-spotted cat (*Prionailurus rubiginosus*) is known to eat rodents [21].

Studies on predation and feeding behavior of leopard cats have been conducted in countries, such as Singapore [7], Malaysia [8], Thailand [9, 10], Russia [11], Indonesia [12], Pakistan [13], India [14], Cambodia [15], Laos [16], Korea [17], and Japan [18, 22]. In these studies, prey items were identified by direct observation of hunting [12], observation of stomach contents [18], and microscopic observation of hair, teeth, feathers, toenails, and bones contained in collected fecal samples [7–11, 13–17]. However, these methods require not only labor and time, but also expertise in morphology and osteology. Furthermore, the identification of closely related prey items is often difficult by morphological observation.

The recent introduction of DNA barcoding in wildlife dietary analysis [23–28] has made it possible to overcome the difficulties described above. In carnivore and omnivore dietary studies [23–28], DNA markers such as the vertebrate mitochondrial 12S rRNA gene [29] have been targeted to analyze DNA derived from diets in collected fecal samples. DNA barcoding has also been applied to dietary analysis of leopard cats in Pakistan [30], China [31, 32], and Korea [33]. In the Korean study [33], bone or tissue fragments were isolated from fecal samples, DNA was extracted from each of the isolates, and each DNA extract was analyzed individually by Sanger sequencing. In the Chinese study [31], DNA was extracted collectively from fecal samples, the DNA markers were amplified and cloned, and each clone was individually analyzed by Sanger sequencing. In the remaining studies [30, 32], high-throughput sequencing (HTS) was used to analyze DNA collectively extracted from feces without cloning nor isolating individual tissue or bone fragments. The HTS-based DNA barcoding is particularly useful for analyzing dietary diversity and detecting rare species due to its sensitivity. For the leopard cat dietary survey, the blocking oligonucleotide has also been developed to suppress the amplification of DNA derived from the leopard cat when performing the vertebrate-specific PCR targeting the mitochondrial 12S rRNA gene [30]. The use of blocking oligonucleotides helps increase the detection sensitivity of prey DNA by suppressing the amplification of the predator DNA.

In this study, we aimed to apply HTS technologies to the dietary analysis of leopard cats in Korea. Using HTS, we analyzed the vertebrate mitochondrial 12S rRNA gene in feces of leopard cats collected in winter in inland areas of Korea. The previously reported blocking oligonucleotide [30] was used to suppress the amplification of leopard cat DNA. In addition to vertebrate mitochondrial 12S rRNA gene sequencing, we also applied shotgun metagenomic sequencing. The use of shotgun metagenomic sequencing allows for detection of not only target organisms (i.e., vertebrates) but also non-target organisms such as invertebrates and plants. We compared the results of vertebrate mitochondrial 12S rRNA gene sequencing and shotgun

metagenomic sequencing, and discussed whether the identified genera were reasonable in light of the Korean fauna.

## 2. Materials and methods

### 2.1. Fecal samples

Previously collected fecal samples of leopard cats for the investigation of zoonotic pathogens [34] and antimicrobial resistance genes [35] were used for dietary analysis in this study. Briefly, a total of twenty-two samples were collected in inland areas of Sejong and Gongju cities in Chungcheongnam-do province, and Daegu and Sangju cities and Goryeong gun in Gyeongsangbuk-do province during February 2019 in Korea (S1 Fig and S1 Table in S1 File). Six samples (L1 and L8–L12) were collected in an area in Gongju city, Chungcheong-nam-do, where a small stream originating from a tractional pond is flowing around and a paddy field is spread out. This area is surrounded by mountains and is adjacent to a reservoir. The sample L2 was collected from a location right next to the aforementioned reservoir. Here too, rice fields are spread out, and it is surrounded by mountains. Five samples (L3–L6 and L13) were collected from a basin of a river flowing through Gongju city. This basin is covered with plants and grass and is surrounded by mountains. Two samples (L7 and L14) were collected around a confluence of the aforementioned river and a small stream in Sejong city, Chungcheongnam-do. There is a wetland right next to this sampling area, which is surrounded by mountains. Eight samples (L15–L22) were collected in Gyeongsang-buk-do province, and one of them (L15) was collected in Sangju city. In the area where the sample L15 was collected, a small stream originating from a tractional pond is flowing around, rice fields are spread out, and it is surrounded by mountains. The remaining seven samples (L16–L22) were collected from a basin of a river flowing through Gyeongsangbuk-do province. As with other sampling sites, there are paddy fields around this place and are surrounded by mountains. No permission was required to collect the fecal samples since they were collected in public places. About 10 g of each sample was collected with a wooden spatula and placed in a 50 ml tube. The samples taken were transported to the laboratory on ice and stored at −80°C for analysis. Our previous study [34] has confirmed that the collected fecal samples are from leopard cats by the leopard cat-specific PCR assay using a primer pair PrioF/PrioR [36].

### 2.2. DNA extraction

DNA extraction from the collected scat samples was performed by a PowerMax® Soil DNA Isolation Kit (Mobio Laboratory, Inc., Carlsbad, CA, USA). First, 5 ml of ultra-pure water was added to each of the 50 ml tubes, and the inner part of the fecal sample was used whenever possible to minimize potential contamination from the environment and homogenized using a wooden spatula [24]. Second, 0.2 g of each homogenized sample, 300 mg of 0.1 mm diameter glass beads, and 100 mg of 0.5 mm diameter glass beads were added into a 2 ml tube of the DNA isolation kit. Third, the samples were additionally homogenized by a bead beater (BioS-pec Products, Inc., Bartlesville, OK, USA) for 3 min. Finally, from each sample, DNA was extracted and purified using the kit's protocol and eluted in 50 μl TE (10 mM Tris-HCl, 1 mM EDTA, pH = 8.0). The eluted DNA extracts were stored at −80°C until analysis.

### 2.3. Shotgun metagenomic sequencing

In this study, we used shotgun metagenomic data previously obtained for the purpose of investigating antimicrobial resistance genes [35]. In the present study, we specifically analyzed

dietary content. Briefly, eleven randomly selected from a total of twenty-two samples were subject to shotgun metagenomic sequencing. Shotgun metagenomic sequencing was performed on an Illumina NextSeq system (Illumina, San Diego, CA, USA) with 150-bp paired-end chemistry. For details, see our previous study [35].

From the resulting shotgun metagenomic sequencing reads, low-quality reads were eliminated and adapter sequences were trimmed using Trimmomatic version 0.39 [36] with default settings. Then, paired-end reads were merged by FLASH version 1.2.11 [37]. Finally, the paired shotgun metagenomic reads were taxonomically annotated against the RefSeq database using BLASTX with default parameters (e-value cutoff = $10^{-5}$, % identity cutoff = 60, and alignment length cutoff = 15 bp) in the Metagenomics Rapid Annotation (MG-RAST) server version 4.0.3 [38]. As of July 2022, the genome sequences of 2,255 vertebrate species have been reported as RefSeq, and these are considered to be contained in the MG-RAST database. Shotgun metagenomic sequence reads included those that are not related to the diet, such as bacteria and fungi. We extracted only sequences assigned to the phyla Chordata, Streptophyta, and Arthropoda, which are thought to be related to the diet of the leopard cat. Also, the reads assigned to the family Felidae to which the leopard cat belongs were excluded because they are likely to come from the leopard cat itself rather than the diet.

## 2.4. Vertebrate mitochondrial 12S rRNA gene sequencing

All twenty-two samples collected were subjected to vertebrate mitochondrial 12S rRNA gene amplicon sequencing. Of them, seven samples were technically duplicated. The libraries were prepared with a primer pair of 12SV5F and 12SV5R that target the vertebrate mitochondrial 12S rRNA gene and a blocking oligonucleotide *PrioB* that inhibits the amplification of leopard cat's DNA [29, 30]. The primers were added with adapter sequences for tagged sequencing of Illumina MiSeq. Each PCR mixture with a volume of 50 µl contained 25 µl of Premix Taq™ (Takara Bio Inc., Otsu, Shiga, Japan), 0.1 µM of each primer, 2 µM of *PrioB*, 13 µl of molecular grade water, and 1 µl of DNA extract. PCR started with initial denaturation at 95˚C for 15 min, followed by 45 cycles of 30 s at 95˚C and 30 s at 60˚C. There was no elongation step [30]. The resulting amplicons were purified with AMPure XP beads (Beckman Coulter, Inc., Brea, CA, USA). The purified amplicons were tagged with a Nextera XT Index kit (Illumina), with PCR starting with initial denaturation at 95˚C for 3 min, followed by 8 cycles of 30 s at 95˚C, 30 s at 55˚C, and 30 s at 72˚C, and completing by final extension at 72˚C for 5 min. The tagged libraries were purified with AMPure XP beads (Beckman Coulter). The purified tagged libraries were quantified by a Quant-iT PicoGreen dsDNA reagent kit (Life Technologies, Carlsbad, CA, USA). The quantified libraries were normalized, pooled, and introduced to a v3 600 cycle-kit reagent cartridge (Illumina) with 30% PhiX control. The loaded libraries were sequenced on an Illumina MiSeq system with 300-bp paired-end chemistry.

From the resulting vertebrate mitochondrial 12S rRNA gene sequence reads, low quality reads with a score of less than 20 were removed and the adapter sequences were trimmed using MiSeq Reporter version 2.5 (Illumina). In addition, ambiguous base callings were trimmed by Trimmomatic v 0.33 [36]. Then, OBITools [39] was used for subsequent processing and analyses. The paired-end reads were concatenated by the *illuminapairedend* command. The concatenated reads were merged and dereplicated by the *obiuniq* command and filtered with a cut-off length of 80 bp by the *obigrep* command. The *obiclean* command was used to exclude erroneous reads. Using the *ecotag* command, the high-quality reads obtained were taxonomically assigned against our custom reference database [28] consisting of vertebrate mitochondrial 12S rRNA gene sequences downloaded from the EMBL nucleotide

database. The database contains reference mitochondrial 12S rRNA gene sequences of a total of 12,574 vertebrate species.

## 2.5. Statistical analysis

Statistical analysis was performed to compare the results of shotgun metagenomic sequencing and vertebrate mitochondrial 12S rRNA gene sequencing. The differences in diversity, structure, and membership of prey animals identified by both methods were compared at the class level. The phyloseq [40] and vegan [41] packages in R version 4.1.0 were used. Prior to the comparison, from both sequencing results, only reads classified as the phylum Chordata were included and the reads identified as the family Felidae were excluded. The *tax_glom* function of phyloseq was used to merge taxa of the same group at a given taxonomic rank. Then, using *rarefy_even_depth* function of phyloseq, each library was rarefied to 2,700 reads, which is the round-down number of the minimum number of the Chordata reads (i.e., 2,744 reads per library) obtained by shotgun metagenomic sequencing (S2 Table in S1 File). The *estimate_richness* function of phyloseq was used to compute α diversity indices of the rarefied libraries, and the Wilcoxon rank-sum test was used to compare of the results obtained by the two sequencing methods. In addition, we compared the differences in prey community structure and membership obtained by the two different sequencing methods. The Bray–Curtis dissimilarity and Jaccard index were used to represent the community structure and membership, respectively. The vegan package was used to perform permutational multivariate analysis of variance (PERMANOVA) to compare the differences. Additionally, the Mantel test was performed based on the Bray–Curtis dissimilarity with 10,000 permutations, in order to evaluate the Spearman's rank correlation between the distance matrices of shotgun metagenomic sequencing and vertebrate mitochondrial 12S rRNA gene sequencing.

## 3. Results

### 3.1. Sequencing statistics

From twenty-two fecal samples, twenty-two vertebrate mitochondrial 12S rRNA gene sequence and eleven shotgun metagenomic sequence libraries were obtained. For vertebrate mitochondrial 12S rRNA gene sequencing, seven additional libraries were obtained as technical replicates. A total of 2,816,069 high-quality reads were obtained by vertebrate mitochondrial 12S rRNA gene sequencing (S3 Table in S1 File). A total of 122,474,517 high-quality shotgun metagenomic sequencing reads were obtained. Among them, a total of 352,045 reads (0.29%) were assigned to the phyla Arthropoda, Chordata, or Streptophyta, which are thought to be derived from the diet of leopard cats (S2 Table in S1 File).

### 3.2. Shotgun metagenomic sequencing

Among the reads assigned to the phyla related to the diet of leopard cats, their relative abundances were 41.7% for Chordata, 47.1% for Streptophyta, and 11.2% for Arthropoda. The relative abundance of diet-related phyla detected in each sample is shown in Fig 1A. At the class level, abundant vertebrates and their relative abundance within the phylum were Mammalia (62.7%), Amphibia (19.3%), Actinopterygii (ray-finned fishes) (7.3%), and Aves (birds) (3.8%) (Fig 1B). Among the Streptophyta, the abundant classes were Magnoliopsida (dicotyledons) (76.3%), Bryopsida (mosses) (12.9%), Liliopsida (monocotyledons) (7.6%), and Isoetopsida (quillworts) (1.2%) (Fig 1C). Among the Arthropoda, the abundant classes were Insecta (94.0%) and Arachnida (spiders, ticks, and mites) (5.5%) (Fig 1D).

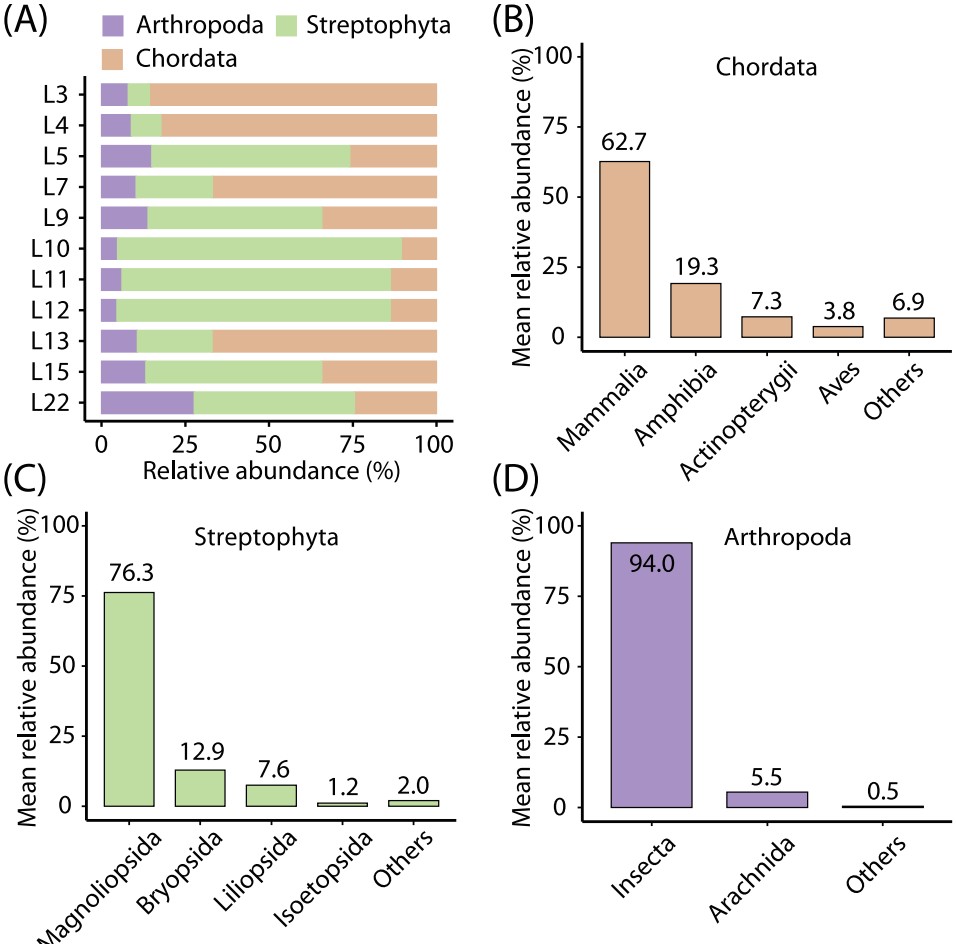

**Fig 1. Shotgun metagenomic sequencing results.** Mean relative abundance of the leopard cat diet-related phyla detected from their fecal samples are shown (n = 11). The reads classified into the family Felidae were excluded because they may be derived from DNA of the leopard cat itself. (A) Relative abundance of the three leopard cat diet-related phyla in each sample. Mean relative abundance of each class within each phylum of (B) Chordata, (C) Streptophyta, and (D) Arthropoda.

### 3.3. Vertebrate mitochondrial 12S rRNA gene sequencing

Fig 2 shows the relative abundance of the 20 most abundant genera detected by vertebrate mitochondrial 12S rRNA gene sequencing. Murids such as *Micromys* (harvest mouse), *Apodemus* (field mouse), and *Rattus* (rat) were abundantly detected. They belong to a group of rodents. Other small-size rodents, such as *Sciurus* (red squirrel) and *Myodes* (vole), were also detected. Relatively large herbivores such as *Hydropotes* (water deer) were also detected. Moreover, carnivores such as *Mustela* (weasel) and *Canis* (domestic dog) were also detected in some samples. Birds, such as *Streptopelia* (dove) and *Phalacrocorax* (cormorant), fishes, such as *Squaliobarbus* (barbel chub) and *Lepomis* (bluegill sunfish), and amphibians, such as *Pelophylax* (true frog), were also detected. Technical duplications show reproducible results (S2 Fig in S1 File). There was no spatial difference in the prey composition between Chungcheongnam-do and Gyeongsangbuk-do, where sampling was carried out (S3 Fig in S1 File).

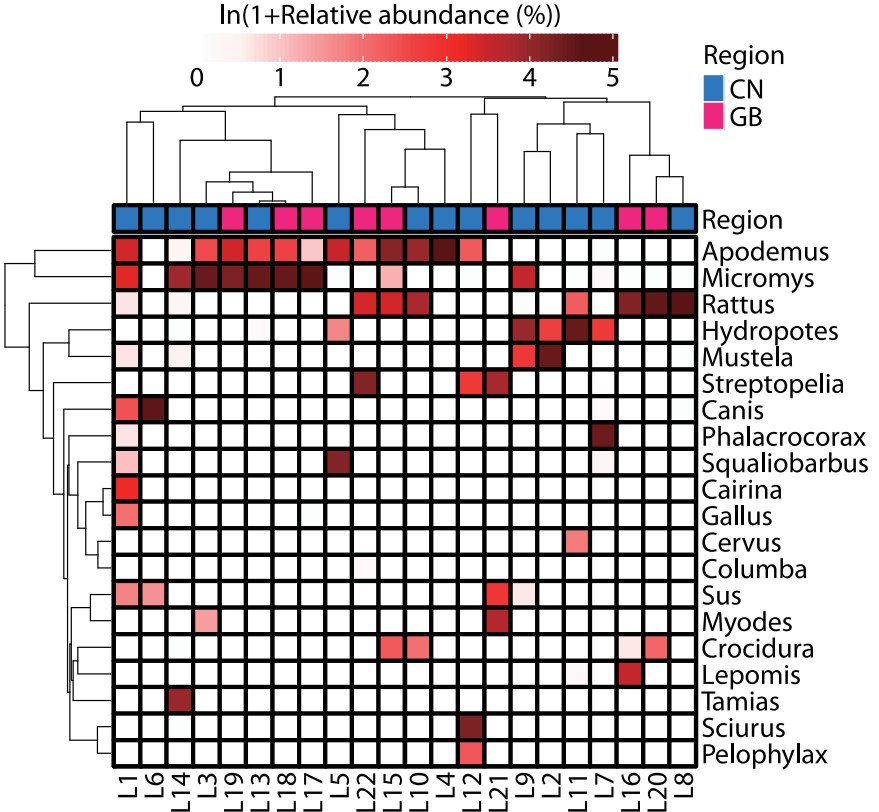

**Fig 2. Vertebrate mitochondrial 12S rRNA gene sequencing results.** Relative abundance of the top 20 genera identified from fecal samples (n = 22) by vertebrate mitochondrial 12S rRNA gene sequencing is shown. The ambiguous reads that were not identified at the genus level and the reads that were assigned to genera belonging to the family Felidae were excluded from the calculation. The dendrograms represent the similarity (dissimilarity) of log-transformed relative abundance results between samples (x-axis) or between prey animals (y-axis). The dissimilarity is represented by branch lengths based on Euclidean distance.. Abbreviation: CN, Chungcheongnam-do; GB, Gyeongsanbuk-do.

## 3.3. Comparison between shotgun metagenomic and vertebrate mitochondrial 12S rRNA gene sequencing

The results obtained by shotgun metagenomic sequencing and vertebrate mitochondrial 12S rRNA gene sequencing were compared at two taxonomic levels, i.e., class and genus levels. The class-level comparisons are shown in Fig 3. The proportion of the class Mammalia was highest by both shotgun metagenomic and vertebrate mitochondrial 12S rRNA gene sequencing (Fig 3A). However, the proportion of mammals is highly distinct by vertebrate mitochondrial 12S rRNA gene sequencing, while more diverse classes such as Amphibia and Actinopterygii were detected more evenly by shotgun metagenomic sequencing. These tendencies were also confirmed by α diversity analysis (Fig 3B). We found that the Chao1 estimator (community richness) and the Shannon index (community diversity) were statistically significantly higher by shotgun metagenomic sequencing than by vertebrate mitochondrial 12S rRNA gene sequencing ($p < 0.01$; Wilcoxon rank-sum test). Moreover, β diversity analysis found that both Bray–Curtis dissimilarity (community structure) and Jaccard index (community membership) differed statistically significantly between shotgun metagenomic and vertebrate mitochondrial 12S rRNA gene sequencing ($p < 0.01$ and 0.001, respectively;

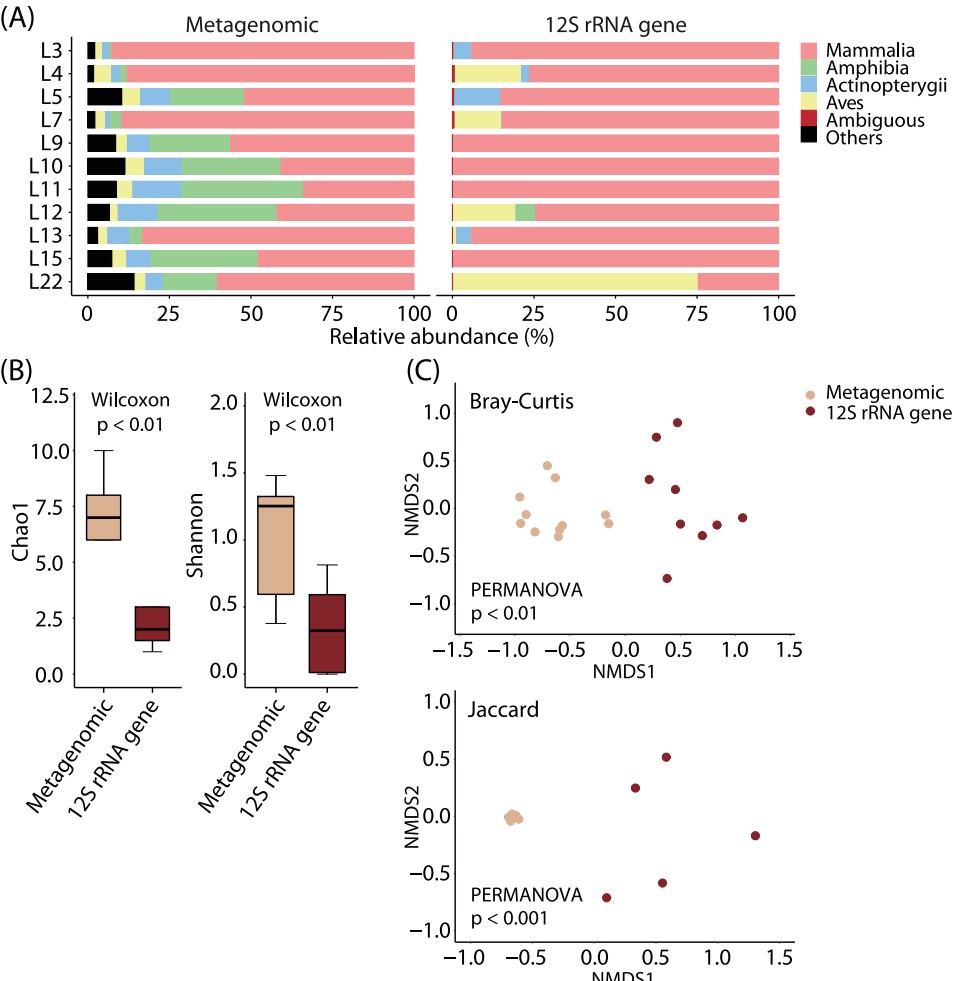

**Fig 3. Comparisons between shotgun metagenomic and vertebrate mitochondrial 12S rRNA gene sequencing.** The class-level results are shown. The reads assigned to the family Felidae were excluded from the comparisons. (A) Relative abundance of classes within the phylum Chordata. (B) Alpha diversity. The Chao1 estimator (community richness) and Shannon index (community diversity) based on the class-level results are shown. (C) Beta diversity. Non-metric multidimensional scaling (NMDS) plots based on the Bray–Curtis dissimilarity (community structure) and Jaccard index (community membership) based on the class-level results are shown.

PERMANOVA) (Fig 3C). No correlation was found between the distance matrices of shotgun metagenomic sequencing and vertebrate mitochondrial 12S rRNA gene sequencing at the class level ($r = -0.05$, $p > 0.05$; Mantel test).

The genus-level comparisons are shown in Table 1, which shows major genera detected by vertebrate mitochondrial 12S rRNA gene and/or shotgun metagenomic sequencing and their inhabitation status in Korea. Most of the major genera detected by vertebrate mitochondrial 12S rRNA gene sequencing, such as *Apodemus* (field mouse), *Micromys* (harvest mouse), and *Hydropotes* (water deer), are known to inhabit Korea. However, many of the major genera identified by shotgun metagenomic sequencing are not known to naturally inhabit Korea. For instance, *Xenopus* (clawed frog) and *Danio* (danio) were abundantly identified. Mammals such as *Pan* (chimpanzee), *Pongo* (orangutan), *Macaca* (macaque), *Ailuropoda* (giant panda), *Monodelphis* (short-tailed opossum), and *Ornithorhynchus* (platypus) were also identified. However, they are not known to live naturally in Korea. Marine chordates such as

**Table 1. Major genera detected and their inhabitation status in Korea.** Twenty most abundant vertebrate and chordate genera detected by vertebrate mitochondrial 12S rRNA gene (n = 11) and/or shotgun metagenomic (n = 11) sequencing are shown.

| Genus | Common name | 12S rRNA gene (%) [a] | Metagenomic (%) [a,b] | Inhabitation in Korea [ref.] |
|---|---|---|---|---|
| *Apodemus* | Field mouse | 25.8 | n.d. | Confirmed [6] |
| *Micromys* | Harvest mouse | 18.9 | n.d. | Confirmed [6] |
| *Hydropotes* | Water deer | 14.3 | n.d. | Confirmed [6] |
| *Rattus* | Rat | 9.7 | 9.1 | Confirmed [6] |
| *Phalacrocorax* | Cormorant | 7.7 | 0.001 | Confirmed [42] |
| *Streptopelia* | Dove | 7.2 | n.d. | Confirmed [43] |
| *Sciurus* | Red squirrel | 6.2 | 0.003 | Confirmed [6] |
| *Squaliobarbus* | Barbel chub | 5.8 | n.d. | Confirmed [44] |
| *Mustela* | Weasel | 1.4 | n.d. | Confirmed [6] |
| *Crocidura* | Shrew | 1.3 | n.d. | Confirmed [6] |
| *Pelophylax* | True frog | 0.8 | n.d. | Confirmed [45] |
| *Cervus* | Deer | 0.5 | 0.003 | Nearly extinct, but raised on farms and very few individuals live in the wild [6, 46–49] |
| *Myodes* | Vole | 0.3 | n.d. | Confirmed [6] |
| *Sus* | Wild boar | 0.1 | 1.1 | Confirmed [6] |
| *Columba* | Pigeon | 0.02 | n.d. | Confirmed [43] |
| *Micropterus* | Largemouth bass | 0.02 | n.d. | Confirmed [50] |
| *Lepomis* | Bluegill sunfish | 0.02 | n.d. | Confirmed [50] |
| *Euryoryzomys* | Rice rat | 0.02 | n.d. | Not confirmed |
| *Canis* | Dog | 0.01 | 17.1 | Pet animal |
| *Xenopus* | Clawed frog | n.d. | 19.3 | Not confirmed |
| *Homo* | Human | n.d. | 8.0 | Confirmed |
| *Danio* | Danio | n.d. | 7.2 | Pet animal |
| *Bos* | Cattle | n.d. | 6.6 | Domestic animal |
| *Mus* | House mouse | n.d. | 5.8 | Confirmed [6] |
| *Branchiostoma* | Lancelet | n.d. | 4.4 | Marine chordate |
| *Ailuropoda* | Giant panda | n.d. | 2.7 | Not confirmed |
| *Ciona* | Sea squirt | n.d. | 2.5 | Marine chordate |
| *Gallus* | Fowl | n.d. | 2.4 | Domestic animal |
| *Pan* | Chimpanzee | n.d. | 2.2 | Not confirmed |
| *Macaca* | Macaque | n.d. | 2.0 | Not confirmed |
| *Pongo* | Orangutan | n.d. | 1.7 | Not confirmed |
| *Monodelphis* | Short-tailed opossum | n.d. | 1.7 | Not confirmed |
| *Taeniopygia* | Finch | n.d. | 1.4 | Pet animal |
| *Equus* | Horse | n.d. | 1.3 | Domestic animal |
| *Ornithorhynchus* | Platypus | n.d. | 1.2 | Not confirmed |
| *Oryctolagus* | Rabbit | n.d. | 0.5 | Pet animal |

[a] The reads assigned to the family Felidae and ambiguous reads at the genus level were excluded from calculation of mean relative abundance.

[b] The relative abundance of each genus was calculated for the total number of reads of identified genera belonging to the phylum Chordata.

Abbreviation: n.d., not detected.

*Branchiostoma* (lancelet) and *Ciona* (sea squirt) were identified, but the detection of these organisms is likely erroneous given that fecal sampling was done inland.

## 4. Discussion

In this study, we applied vertebrate mitochondrial 12S rRNA gene sequencing and shotgun metagenomic sequencing to investigate the dietary content of leopard cats inhabiting inland areas of Korea in winter. We found that the leopard cats fed mainly on mammals, especially murids, and confirmed that the detected prey items were plausible in light of the Korean fauna. In addition, shotgun metagenomic sequencing revealed that the amount of plant DNA in the feces of leopard cats is comparable to that of animal DNA. Furthermore, some inconsistencies between vertebrate mitochondrial 12S rRNA gene sequencing and shotgun metagenomic sequencing, and misidentification of prey at the genus level by shotgun metagenomic sequencing were identified. Below, we discuss these points.

### 4.1. Mammals as main prey for leopard cats in Korea

Both vertebrate mitochondrial 12S rRNA gene sequencing and shotgun metagenomic sequencing revealed that mammals are the main prey of leopard cats in our study areas (Fig 3A). In particular, vertebrate mitochondrial 12S rRNA gene sequencing revealed murids, such as *Micromys*, *Apodemus*, and *Rattus*, and other small-size rodents, such as *Sciurus* and *Myodes*, as the predominant prey (Fig 2 and Table 1). Our observation was the same as that by previous Korean studies [17, 33] that reported these three murid genera and rodents as the main prey of leopard cats. However, this tendency is universal not only for leopard cats in Korea but also for those in other countries [7–17, 22, 30–33].

Large mammals, such as *Sus* and *Hydropotes*, were also detected by vertebrate mitochondrial 12S rRNA gene sequencing. *Hydropotes inermis* (water deer) and *Sus scrofa* (wild boar) grow up to 15 kg [51] and 300 kg [52], respectively, while the leopard cat grows only up to 7.1 kg [1]. Due to the size difference, it can be thought that hunting is not easy, if not impossible. Wild boar and deer, such as *Cervus nippon* (sika deer) and *Capreolus pygargus* (roe deer), were reported to be detected from the feces of leopard cats in the Russian Far East [11] and in Pakistan [13], too. These studies suggest that leopard cats may have scavenged the carcasses of these large animals, although there is no direct evidence presented. In addition to these studies, consumption of deer by hunting or scavenging by leopard cats has been reported, e.g., *Tragulus javanicus* (lesser mouse deer) in Thailand [9] and *Muntiacus* spp. (muntjac) in Cambodia [15] and Laos [16]

Carnivorous animals, such as *Canis* and *Mustela*, were also detected by vertebrate mitochondrial 12S rRNA gene sequencing. In Korea, the wolf is thought to be extinct [6]. Therefore, *Canis* detected is likely domestic dogs. In Pakistan, it has been reported that leopard cats prey on domestic dogs [13]. *Mustela nivalis* (least weasel) and *Mustela sibirica* (Siberian weasel) are known to inhabit Korea [6], and *M. sibirica* was reported to be detected from a feces of leopard cat in Korea [17]. Although it is unclear whether it is predation or scavenging, it has been reported that carnivorous animals were detected in the feces of leopard cats, e.g., *Nyctereutes procyonoides* (raccoon dog) in the Russian Far East [11], *Urva auropunctata* (small Indian mongoose) and *Herpestes edwardsii* (grey mongoose) in Pakistan [13], and civet in Cambodia [15].

We detected an OTU with a sequence highly similar to that of *Cervus* (Fig 2 and Table 1), which is believed to be extinct in the wild in South Korea [6]. The detected sequence is phylogenetically distant from those of *Hydropotes*, an extant deer species in Korea, so the misidentification of *Hydropotes* is unlikely (S4 and S5 Figs in S1 File). We consider that the detected

sequence is from *Cervus*, as they may still exist in South Korea. In fact, species such as *Cervus nippon* (sika deer), *Cervus elaphus* (red deer), and *Cervus canadensis* (elk) are imported from abroad and farmed for their antlers and blood [46, 47]. It has been also reported that the sika deer has been released into the wild by religious events, and about 50 to 100 individuals are known to inhabit Mt. Songni near Chungcheongbuk-do and Gyeongsangbuk-do [6, 48]. There have also been reports of some deer escaping farmland or being illegally released by farmers in Korea [49]. Therefore, albeit in very low numbers, there is a possibility that deer belonging to *Cervus* inhabit the wild environment of Korea, and it is possible that they were hunted by leopard cats.

The genus *Euryoryzomys* detected (Table 1) is not known to inhabit Korea. We consider that the detection of *Euryoryzomys* was a misidentification of *Myodes* because these two rodents have very similar vertebrate mitochondrial 12S rRNA gene sequences (S6 Fig in S1 File). One hypothesis is that the intra-genus variability in the vertebrate mitochondrial 16S gene sequences within the genus *Myodes* may be comparably large to the inter-genus variability between *Myodes* and *Euryoryzomys*. However, this needs to be clarified in future research.

## 4.2. Non-mammalian vertebrates and invertebrates as secondary prey

This study found that mammals are the main prey for the leopard cats in Korea, which may be a universal tendency worldwide. However, exceptions have been also reported, such as leopard cats inhabiting the subtropical Iriomotejima Island in Japan, whose main prey is frogs [18]. Therefore, there may be the regional differences, but their adaptability may allow them to flexibly change their target prey depending on the type of prey locally available. In addition to regional effects, there may also be seasonal effects [13, 15, 17]. In Korea, which is located in the temperate zone, poikilotherms, such as amphibians, hibernate in the cold winter, so they may be inevitably excluded from the predation of leopard cats. However, we found that amphibians, including *Pelophylax* (Fig 2 and Table 1), were present as prey for leopard cats in a relatively high proportion (Fig 1B), even though our samples were collected in the cold winter. A previous study has also reported that the proportion of amphibians in feces of leopard cats in Korea is higher in winter than in other seasons [17]. We do not know the reason. One possible explanation is that the activity of poikilotherms declined in the cold winter, making them easy targets for leopard cats. However, future seasonal investigations need to confirm the tendency and reveal the cause behind.

In addition to Amphibia, Actinopterygii and Aves were relatively abundantly detected as non-mammalian vertebrates by shotgun metagenomic sequencing (Figs 1B and 3A). Vertebrate mitochondrial 12S rRNA gene sequencing also identified genera belonging to Actinopterygii, such as *Squaliobarbus* and *Lepomis*, and Aves, such as *Streptopelia* and *Phalacrocorax* (Fig 2 and Table 1). The detection of fishes makes sense because fecal samples were collected near rivers and reservoirs. Moreover, the detection of the abovementioned fish genera is reasonable as species belonging to these genera have been reported to inhabit Korea, such as *Squaliobarbus curriculus* (barbel chub) [44], *Lepomis macrochirus* (bluegill sunfish) [50], *Streptopelia orientalis* (oriental turtle dove) [43], and *Phalacrocorax carbo* (great cormorant) [42]. In countries other than Korea, several studies have reported that leopard cats eat fish [11, 18, 22, 30, 32] and birds [7–16, 22, 30–32]. In Korea, studies reported that leopard cats eat fish, such as *Monopterus albus* (Asian swamp eel) [33], and birds, such as *Phasianus colchicus* (common pheasant) [33], *Turdus pallidus* (pale thrush) [33], and *Garrulus glandarius* (Eurasian jay) [17]. In this study, we also confirmed that leopard cats eat fish and birds, although they are less common than mammals.

Arthropods were also detected in relatively large numbers by shotgun metagenomic sequencing, most of which were found to be insects (Fig 1D). Many studies have reported that leopard cats feed on arthropods such as insects [7–9, 11, 13, 15, 17, 22] and arachnids [8]. In Korea, cicadas and grasshopper are reported to be detected from feces of leopard cats [17]. It was remarkable that relatively abundant insects were detected from the leopard cat feces collected in the winter in this study, as insects are inactive in the cold winter in Korea. The detection of insects in winter has also been reported in the previous Korean study [17]. The reason is unknown, but we speculate that leopard cats may feed on insect larvae and/or nymphs in winter. In addition, in this study using shotgun metagenomic sequencing, the types of insects detected are unknown. In the future, target-specific sequencing, such as invertebrate 16S rRNA gene sequencing [23], will need to reveal the species of insects that leopard cats specifically prey on.

### 4.3. Large amount of plant DNA in feces of leopard cats

Shotgun metagenomic sequencing revealed that the amount of plant DNA in the feces of leopard cats is comparable to the amount of animal DNA (Fig 1A). Although the amount of DNA is not necessarily proportional to the amount of biomass, it is remarkable that based on the amount of DNA, the amount of plants is comparable to the amount of animals in the feces of leopard cats. The leopard cat is known to eat grass. They are thought to eat grass for intestinal regulation [22] and parasite removal [13], and/or ingestion of minerals and vitamins [17]. In fact, it has been reported that graminoids (grass-like plants, mostly monocot plants) were detected in the feces of leopard cats [17, 22]. However, the majority of plants detected in this study were Magnoliopsida (Fig 1C), which are dicot plants. Therefore, the plants detected in this study do not appear to be consistent with the general dietary habit of leopard cats. One explanation is that graminoids might have become scarce at the time of our fecal sampling done in winter. Meanwhile, the plants belonging to the class Magnoliopsida such as *Bellis perennis* (daisy) were also reported to have been detected abundantly from feces of leopard cats, and the possibility that they were ingested secondarily by eating herbivorous prey was suggested [13]. Another study also reported the detection of dicotyledonous plants such as the genus *Solanum* and subfamily Rosoideae in the feces of leopard cats [31]. These suggest that leopard cats may flexibly adapt to seasonally and locally available plants. Note that similar to the vertebrate results, the plants identified at the genus level by shotgun metagenomic sequencing are likely biased toward species whose genomes have already been sequenced (S4 Table in S1 File). Therefore, to precisely identify plant taxa that were consumed or secondarily ingested by leopard cats, future research should perform sequencing of plant DNA markers (e.g., ITS) for which a broader range of plant taxa are covered in reference databases.

### 4.4. Inaccurate genus-level prey identification by shotgun metagenomic sequencing

In this study, we explored the possibility for shotgun metagenomic sequencing to identify the diet of leopard cats at the genus level (Table 1). In the light of the Korean fauna, we conclude that many of the genera identified by shotgun metagenomic sequencing are erroneous. While many genera, except *Euryoryzomys*, identified by vertebrate mitochondrial 12S rRNA gene sequencing are known to inhabit Korea and reasonable in light of the known eating habits of leopard cats, most of the genera identified by shotgun metagenomic sequencing are not reasonable. For example, the inhabitation of *Xenopus*, *Danio*, *Ailuropoda*, *Pan*, *Macaca*, *Pongo*, *Monodelphis*, and *Ornithorhynchus* in Korea is unknown. Moreover, the detection of marine

organisms such as *Branchiostoma* and *Ciona* is not reasonable because the samples were collected inland far from the sea.

The plausible explanations for why taxonomic assignment of plants and animals by shotgun metagenomic sequencing is inaccurate are given by Pearman et al. [53]. As one reason, they pointed out that plant and animal genome databases are far more incomplete than those of bacteria and fungi. We also consider that this is the reason why taxonomic assignment by shotgun metagenomic sequencing is less accurate than by vertebrate mitochondrial 12S rRNA gene sequencing. In fact, the reference database for shotgun metagenomic sequencing used in this study contained genomic sequences of only 2,255 chordate species, which is much less than sequences of 12,574 vertebrate species contained in the reference database used for vertebrate mitochondrial 12S rRNA gene sequencing. As a result, many reads are presumed to have been biasedly misassigned to organisms whose whole genome sequences have been elucidated, such as *Xenopus tropicalis* [54], *Danio rerio* [55], *Bos taurus* [56], *Branchiostoma floridae* [57], *Ciona intestinalis* [58], *Macaca mulatta* [59], *Monodelphis domestica* [60], and *Taeniopygia guttata* [61]. However, we expect this problem will be alleviated in the future by the ongoing efforts made to expand genomic databases, such as the Earth BioGenome Project [62] and the Genome 10K [63].

### 4.5. Sequencing method biases

We used shotgun metagenomic and vertebrate mitochondrial 12S rRNA gene sequencing, and found the differences in taxonomic richness, and community structure and membership of prey between these two sequencing methods (Fig 3). Specifically, we found that more diverse taxa were detected by shotgun metagenomic sequencing than by vertebrate mitochondrial 12S rRNA gene amplicon sequencing (Fig 3B), and the similar tendency has been reported by previous studies [64–66]. This may be due to the difference in two sequencing approaches, i.e., the presence or absence of targeted amplification process. Bias that can occur during PCR process are caused by various factors such as temperature ramp rates [67], amplicon length, and PCR primer [68, 69]. As an example of bias, rare taxa are known to be underestimated during PCR process [70]. Therefore, the smaller number of taxa detected by vertebrate mitochondrial 12S rRNA gene sequencing in this study is plausible and may have been associated with the inclusion of PCR amplification in the process of library preparation. Additionally, we found that there are differences in community structure and membership between these two sequencing methods (Fig 3C). In addition to the abovementioned amplification bias, the bias due to the difference in the number of copies of the target DNA marker to be amplified [71] may contribute to the observed differences.

In contrast, other previous study [72] reported that the results of amplicon-based sequencing show more diverse taxa than those of shotgun metagenomic sequencing. One reason could be related to inadequate sequencing depth [73]. In our study, only about 0.29% of all shotgun metagenomic reads were assigned to taxa related to prey of leopard cats. This underscores the importance of sequencing depth in wildlife dietary analysis by the shotgun metagenomic approach.

### 4.6. Caveats

In this study, we used DNA sequencing to conduct a dietary survey of leopard cats. We recognize that quantified DNA does not necessarily represent the accurate proportion of biomass consumed, as correctly pointed out by previous studies [74–76]. As discussed above, DNA sequencing can be affected by a variety of factors, such as genome size, copy number variation, PCR bias, primer bias, and so on. Additionally, the difference in digestibility of dietary items

may also affect the relative abundance of DNA measured. However, DNA sequencing can objectively identify dietary items without morphological and osteological expertise and detect rare dietary items. We believe that DNA sequencing helps complement existing methods due to its convenience, objectivity, and detection sensitivity.

## 5. Conclusion

The combination of vertebrate mitochondrial 12S rRNA gene amplicon sequencing and shotgun metagenomic sequencing provided a comprehensive understanding of the dietary composition of the leopard cat, an endangered species in Korea. By using shotgun metagenomic sequencing, we were able to grasp an overall picture of the dietary composition of leopard cats, including the presence of plants in a relatively large proportion. Both shotgun metagenomic sequencing and vertebrate mitochondrial 12S rRNA gene sequencing confirmed that mammals are the main prey. It was also confirmed that the genera of prey identified by vertebrate mitochondrial 12S rRNA gene sequencing were reasonable in light of the Korean fauna. However, the genera identified by shotgun metagenomic sequencing were inaccurate, which is likely due to the inadequate genomic database, and therefore care must be taken when using shotgun metagenomic sequencing to identify prey at lower taxonomic levels, such as species and genus levels. Meanwhile, the genome database is expanding, and it is expected that this problem will be alleviated in the future. We expect that the use of shotgun metagenomic sequencing will provide a novel opportunity to comprehensively understand the inter-phylum and inter-class dietary composition of wildlife in the near future.

## Supporting information

**S1 File.**
(PDF)

## Author Contributions

**Conceptualization:** Kyung Yeon Eo, Woo-Shin Lee, Junpei Kimura, Naomichi Yamamoto.

**Data curation:** Cheolwoon Woo, Priyanka Kumari.

**Formal analysis:** Cheolwoon Woo, Priyanka Kumari.

**Funding acquisition:** Naomichi Yamamoto.

**Investigation:** Cheolwoon Woo, Priyanka Kumari, Kyung Yeon Eo, Junpei Kimura, Naomichi Yamamoto.

**Project administration:** Naomichi Yamamoto.

**Resources:** Kyung Yeon Eo, Woo-Shin Lee, Junpei Kimura.

**Supervision:** Kyung Yeon Eo, Woo-Shin Lee, Junpei Kimura.

**Validation:** Naomichi Yamamoto.

**Visualization:** Cheolwoon Woo, Naomichi Yamamoto.

**Writing – original draft:** Cheolwoon Woo.

**Writing – review & editing:** Priyanka Kumari, Kyung Yeon Eo, Woo-Shin Lee, Junpei Kimura, Naomichi Yamamoto.

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
