## [Decision Letter · Decision Letter 0]

27 Oct 2022

PONE-D-22-22486Combining vertebrate 12S rRNA gene sequencing and metagenomic sequencing to investigate the diet of the leopard cat (Prionailurus bengalensis) in KoreaPLOS ONE

Dear Dr. Yamamoto,

Thank you for submitting your manuscript to PLOS ONE. After careful consideration, we feel that it has merit but does not fully meet PLOS ONE’s publication criteria as it currently stands. Therefore, we invite you to submit a revised version of the manuscript that comprehensively addresses the points raised during the review process.

We look forward to receiving your revised manuscript.

Kind regards,

Michael Schubert

Academic Editor

PLOS ONE

2. We noted in your submission details that a portion of your manuscript may have been presented or published elsewhere. [DETAILS AS NEEDED] Please clarify whether this [conference proceeding or publication] was peer-reviewed and formally published. If this work was previously peer-reviewed and published, in the cover letter please provide the reason that this work does not constitute dual publication and should be included in the current manuscript.

Reviewers' comments:

Reviewer's Responses to Questions

**Comments to the Author**

1. Is the manuscript technically sound, and do the data support the conclusions?

Reviewer #1: Yes

Reviewer #2: Yes

2. Has the statistical analysis been performed appropriately and rigorously? 

Reviewer #1: Yes

Reviewer #2: Yes

3. Have the authors made all data underlying the findings in their manuscript fully available?

Reviewer #1: Yes

Reviewer #2: Yes

4. Is the manuscript presented in an intelligible fashion and written in standard English?

Reviewer #1: Yes

Reviewer #2: Yes

5. Review Comments to the Author

Reviewer #1: In this manuscript, Woo al. performed 12S rDNA metagenomic sequencing and WGS metagenomic sequencing in fecal samples collected from the wild to determine the dietary content and prey of the endangered leopard cat Prionailurus bengalensis. The 12S rDNA ampliconic sequencing identified many vertebrate species that are naturally found in the habitat, which provide insights into the dietary prey of leopard cats.

However, the results do not agree completely with the results from WGS metagenomic data, which identified DNA from many model organisms, such as zebra fish, grey short-tailed opossum, Xenopus etc, … The authors concluded that WGS metagenomic sequencing is less robust than 12S rDNA sequencing in the genus-level identification of vertebrate prey of leopard cats.

The two major issues are:

The authors mentioned “vertebrate 12S rRNA gene sequencing” and HTS (high-throughput sequencing). Since the authors are sequencing mitochondrial 12S rRNA genes from multiple species using degenerate primers, I would recommend referring the approach as “mitochondrial rDNA metagenomic sequencing”, which is similar to the widely used 16S metagenomic sequencing to study the gut microbiome. I also recommend the authors spell out the metagenomic sequencing as “whole-genome shotgun (WGS) metagenomic sequencing”.

I agree with the authors that results from 12S ampliconic data is a much better representation of the dietary composition. However, I disagree with the discussion that WGS metagenomic sequencing is not robust compared to the 12S approach. Please remove discussion 4.4 because the amplification and copy number differences are not the major cause of inconsistent results between 12S and WGS. I also suggest the authors revise Discussion 4.5. The inaccurate results are not due to the WGS method, but the inadequate annotation of the vertebrate genomes of the leopard cat prey. This is not a sequencing method bias (sequencing method bias exists but it is not the main reason), but the reference bias or annotation bias. The 12S rDNA database consists of over 12,000 species, which provides a decent coverage of the prey. In contrast, MG-RAST vertebrate reference database is heavily biased toward sequenced genomes, and this is why model organisms are enriched in the final annotation. As the authors mentioned, currently, comprehensive genome assemblies of all relevant species are lacking, resulting in inaccurate annotation of the WGS metagenomic reads. However, this has nothing to do with the WGS approach itself. If analyzed properly, WGS would be an accurate representation of the dietary content (comparable to the 12S results), but that would require the availability of 12000 vertebrate genomes.

Please find specific minor comments below:

Line 75, Is there any procedure to minimize environmental contamination during sample collection? (for example, using a fecal loop/spoon to get into the inside of the feces)

Line 229, Figure 2. Do fecal samples collected from the same location have more similar fecal microbiome? I suggest the authors group the 22 fecal samples by their location.

The leopard cat is an endangered species. Are any two samples (for example, L18 and L19) from the same individual? The authors may be able to check this using the leopard reads in the WGS metagenomic sequencing.

What is the percentage of leopard cat DNA for each sample in metagenomic sequencing?

Reviewer #2: Overall, this is an interesting and well written proof-of-concept paper that demonstrates the potential utility of different types of NGS for diet analysis of carnivores. I have suggested a major revision simply because I have some questions regarding the bioinformatic and statistical methods, but these are not likely to substantially alter the results. Specific comments:

Introduction: I would like to see more background on the feeding ecology of leopard cats and closely related species in the intro. Also, please provide some more (non-leopard cat) examples of 12S and metagenomic diet analysis from other carnivore species. Did they encounter similar issues with metagenomic classification?

L75-94: Did you confirm that these samples were from different individuals?

L95-97: How did you ensure that no contamination occurred from soil or the surrounding vegetation where the feces were found?

L124: Did you try adjusting the default e-value or percent identity cutoff values to reduce the number of inaccurate classifications?

L244-249: It would be interesting to know whether there was any correlation between the 12S results and metagenomic results within each individual, i.e. using a mantel test or similar multivariate test.

L188: please provide accession information for the metagenomic information, not just that it was published in a previous study.

L264: Would adjusting the filtering parameters in your bioinformatics pipeline eliminate the presumably erroneous classification of Cervus? Otherwise, what could explain this inaccuracy in the 12S data?

L321-335: this paragraph would be better placed in the next section (non-mammalian vertebrates & invertebrates)

L380: please elaborate on the idea that plants were ingested secondarily by eating herbivorous prey. This is an interesting idea, is there any evidence in other studies that metagenomic sequencing of carnivore diets can detect plant DNA ingested by prey? What type of plants do the prey of leopard cats consume, and did you see any of those plant taxa in the metagenomic data?

L420-423: see earlier comment on BLASTX parameters – would adjusting these parameters reduce the number of misclassifications?

Fig 1A: Rather than using a minimally-informative pie chart, I think it would be better to show a stacked bar chart with the diet composition of each individual sequenced (similar to Fig 3A)

Fig 2: If I understand correctly, the x-axis on this figure shows phylogenetic relationships among taxa. Please include how this tree was generated in the methods section, or elaborate what is meant by "euclidean distance" in the figure caption.

6. PLOS authors have the option to publish the peer review history of their article (what does this mean?). If published, this will include your full peer review and any attached files.

Reviewer #1: No

Reviewer #2: No

---

## [Author Response · Author response to Decision Letter 0]

25 Nov 2022

Please find our responses to the reviewers’ comments below. The parts changed are marked in the revised manuscript, and the line numbers in our responses refer to the revised manuscript with track changes.

Reviewer #1:

1. In this manuscript, Woo al. performed 12S rDNA metagenomic sequencing and WGS metagenomic sequencing in fecal samples collected from the wild to determine the dietary content and prey of the endangered leopard cat Prionailurus bengalensis. The 12S rDNA ampliconic sequencing identified many vertebrate species that are naturally found in the habitat, which provide insights into the dietary prey of leopard cats.

However, the results do not agree completely with the results from WGS metagenomic data, which identified DNA from many model organisms, such as zebra fish, grey short-tailed opossum, Xenopus etc, … The authors concluded that WGS metagenomic sequencing is less robust than 12S rDNA sequencing in the genus-level identification of vertebrate prey of leopard cats.

RESPONSE: Thank you for your interest in our submitted paper and your time to review our paper thoroughly. Your comments have allowed us to improve the paper to a higher quality. In particular, we would like to thank you for suggesting the use of appropriate terminology and for suggesting corrections to the hasty conclusion in our Discussion section.

2. The two major issues are:

The authors mentioned “vertebrate 12S rRNA gene sequencing” and HTS (high-throughput sequencing). Since the authors are sequencing mitochondrial 12S rRNA genes from multiple species using degenerate primers, I would recommend referring the approach as “mitochondrial rDNA metagenomic sequencing”, which is similar to the widely used 16S metagenomic sequencing to study the gut microbiome. I also recommend the authors spell out the metagenomic sequencing as “whole-genome shotgun (WGS) metagenomic sequencing”.

RESPONSE: Thank you for your suggestion. Based on your comments, “vertebrate 12S rRNA gene sequencing” has been changed to “vertebrate mitochondrial 12S rRNA gene sequencing”, and “metagenomic sequencing” has been changed to “shotgun metagenomic sequencing”. The words were changed throughout the manuscript, including the title. 

We prefer using the term “vertebrate mitochondrial 12S rRNA gene sequencing” rather than “vertebrate mitochondrial 12S rDNA metagenomic sequencing”, since vertebrate mitochondrial 12S rRNA gene sequencing is not metagenomic as it targets only a single gene (not genome). We also prefer using the term “shotgun metagenomic sequencing” rather than “whole-genome shotgun (WGS) metagenomic sequencing”, as the latter term contains the similar words “whole-genome” and “metagenomic” redundantly. Metagenomic sequencing intrinsically analyzes whole-genomes of multiple organisms in a given environmental sample. 

3. I agree with the authors that results from 12S ampliconic data is a much better representation of the dietary composition. However, I disagree with the discussion that WGS metagenomic sequencing is not robust compared to the 12S approach. Please remove discussion 4.4 because the amplification and copy number differences are not the major cause of inconsistent results between 12S and WGS. I also suggest the authors revise Discussion 4.5. The inaccurate results are not due to the WGS method, but the inadequate annotation of the vertebrate genomes of the leopard cat prey. This is not a sequencing method bias (sequencing method bias exists but it is not the main reason), but the reference bias or annotation bias. The 12S rDNA database consists of over 12,000 species, which provides a decent coverage of the prey. In contrast, MG-RAST vertebrate reference database is heavily biased toward sequenced genomes, and this is why model organisms are enriched in the final annotation. As the authors mentioned, currently, comprehensive genome assemblies of all relevant species are lacking, resulting in inaccurate annotation of the WGS metagenomic reads. However, this has nothing to do with the WGS approach itself. If analyzed properly, WGS would be an accurate representation of the dietary content (comparable to the 12S results), but that would require the availability of 12000 vertebrate genomes.

RESPONSE: Thank you for your suggestion. As you pointed out, we also agree that the difference between vertebrate mitochondrial 12S rRNA gene sequencing and shotgun metagenomic sequencing is that MG-RAST vertebrate reference database is heavily biased toward already sequenced genome. However, we would like to avoid deleting Discussion 4.4. As shown in Figure 3, we identified a sequencing method bias. This trend has been identified in previous studies as discussed in Discussion 4.4. As you pointed out, the sequencing method bias is highly likely a minor cause, not a major cause. However, we think it is important to provide readers with a broader view in comparing and analyzing the results of vertebrate mitochondrial 12S rRNA gene sequencing and shotgun metagenomic sequencing. As a compromise, we would like to propose the following modifications:

1) Delete the sentence “Our finding suggests that metagenomic sequencing is less robust than 12S rRNA gene sequencing in the genus-level identification of vertebrate prey of leopard cats” 

2) By switching the order of chapters 4.4 and 4.5, we emphasize that the content of previous chapter 4.5 (database bias) is more important.

4. Please find specific minor comments below:

Line 75, Is there any procedure to minimize environmental contamination during sample collection? (for example, using a fecal loop/spoon to get into the inside of the feces)

RESPONSE: Thank you for paying attention to details that we have not covered in the manuscript. We extracted DNA from the inner portion of fecal samples whenever possible to avoid contaminants from the outer surface of the feces. The following sentence has been added: 

“and the inner part of the fecal sample was used whenever possible to minimize potential contamination from the environment” Lines 113-114

5. Line 229, Figure 2. Do fecal samples collected from the same location have more similar fecal microbiome? I suggest the authors group the 22 fecal samples by their location.

RESPONSE: Thank you for your suggestion. Our samples were taken from two regions, Chungcheongnam-do and Gyeongsangbuk-do, and the number of samples obtained from each region was 14 and 8, respectively. By dividing the samples into these two regions, we revised Figure 2 (see our revised manuscript). In addition, we checked whether alpha and beta diversities differed by regions, and this was added as S3 Fig. We did not find significant difference in prey composition by site. On this, the following statement was added.

“There was no spatial difference in the prey composition between Chungcheongnam-do and Gyeongsangbuk-do, where sampling was carried out (S3 Fig in S1 File).” Lines 240-242

6. The leopard cat is an endangered species. Are any two samples (for example, L18 and L19) from the same individual? The authors may be able to check this using the leopard reads in the WGS metagenomic sequencing.

RESPONSE: Thank you for your question. We tried to determine whether the samples used for shotgun metagenomic sequencing were derived from the same individual. However, due to limited recovery and shorter size of the shotgun metagenomic sequencing reads, it is not feasible to assemble these reads into larger genomic contigs to perform such analysis.

7. What is the percentage of leopard cat DNA for each sample in metagenomic sequencing?

RESPONSE: The shotgun metagenomic sequencing reads assigned to the Felidae, family to which leopard cat belongs, varied from 0.0004 – 3.1% of all eukaryotic reads. A lower percentage and higher variability of leopard cat sequences could be due to that these feces were collected sometime after defecation and may have a lower load of host tissues compared to fresh feces. However, as mentioned in the manuscript, each fecal sample was verified for the host (leopard cat) identity by leopard cat-specific PCR assay.

Reviewer #2:

1. Overall, this is an interesting and well written proof-of-concept paper that demonstrates the potential utility of different types of NGS for diet analysis of carnivores. I have suggested a major revision simply because I have some questions regarding the bioinformatic and statistical methods, but these are not likely to substantially alter the results. 

RESPONSE: Thank you for your interest in our submitted paper and your time to review our paper thoroughly. Your comments have allowed us to improve the paper to a higher quality. In particular, your questions about bioinformatics and statistical methods allowed us to check the methods we used once more. 

2. Specific comments:

Introduction: I would like to see more background on the feeding ecology of leopard cats and closely related species in the intro. Also, please provide some more (non-leopard cat) examples of 12S and metagenomic diet analysis from other carnivore species. Did they encounter similar issues with metagenomic classification?

RESPONSE: Thank you for your suggestion. However, we could not easily find previous studies that analyzed the diet of carnivores by applying shotgun metagenomic sequencing. Instead, as you suggested, we added more background on the feeding ecology of closely related species in the Introduction section as below. 

“There is also some literature on the diet of relative species of the leopard cat. For example, the flat-headed cat (Prionailurus planiceps) is known to eat fish, but also shrimp, birds, small rodents, and domestic poultry [1]. The fishing cat (Prionailurus viverrinus) is a species regarded as a generalist and known to feed on a variety of prey including rodents, birds, and fish [2]. The rusty-spotted cat (Prionailurus rubiginosus) is known to eat rodents [3].” Lines 36-41

3. L75-94: Did you confirm that these samples were from different individuals?

RESPONSE: Thank you for your question. We tried to determine whether the samples used for shotgun metagenomic sequencing were derived from the same individual. However, due to limited recovery and shorter size of the shotgun metagenomic sequencing reads, it is not feasible to assemble these reads into larger genomic contigs to perform such analysis.

4. L95-97: How did you ensure that no contamination occurred from soil or the surrounding vegetation where the feces were found?

RESPONSE: Thank you for paying attention to details that we have not covered in the manuscript. Admittedly, it has been difficult to completely eliminate contamination. However, we made efforts to minimize contamination by using the interior of fecal samples whenever possible. The following sentence was added:

“and the inner part of the fecal sample was used whenever possible to minimize potential contamination from the environment” Lines 113-114

5. L124: Did you try adjusting the default e-value or percent identity cutoff values to reduce the number of inaccurate classifications?

RESPONSE: Thank you for your question. The default e-value and/or percent identity cutoff values have not been adjusted. We haven’t tried using e-values higher than the default (1e-5) or lower percent identity cutoff values lower than the default (60%) because it is undesirable [4]. If our understanding is correct, your question is asking whether we tried smaller e-values and/or higher percent identity cutoff values than the default settings in shotgun metagenomic sequencing. 

Even though we applied a smaller e-value or higher percent identity cutoff value, it is expected to produce similar results, i.e., inaccurate classification. We think so because the MG-RAST reference database we used (RefSeq database) is thought to be heavily biased towards the already sequenced genome of 2,255 vertebrate species. The basis for our assumption is that the top-hit are almost model organisms in the results of shotgun metagenomic sequencing. Therefore, we thought that adjusting the e-value or percent identity cutoff value will not help reduce misclassification.

6. L244-249: It would be interesting to know whether there was any correlation between the 12S results and metagenomic results within each individual, i.e. using a mantel test or similar multivariate test.

RESPONSE: Thank you for your suggestion. If we understand your suggestion correctly, your suggestion is to examine the leopard cat individual from which each fecal sample was derived, and then compare the results of vertebrate mitochondrial 12S rRNA gene sequencing and shotgun metagenomic sequencing within each leopard cat individual.

As answered in your third question, we tried to determine whether the samples used for shotgun metagenomic sequencing were derived from the same individual. However, due to limited recovery and shorter size of the shotgun metagenomic sequencing reads, it is not feasible to assemble these reads into larger genomic contigs to perform such analysis.

7. L188: please provide accession information for the metagenomic information, not just that it was published in a previous study.

RESPONSE: Thank you for your suggestion. The following sentence was added: 

“Raw data of shotgun metagenomic sequencing has been published our previous study [5] under the BioProject accession number PRJNA772888.” Lines 198-199

8. L264: Would adjusting the filtering parameters in your bioinformatics pipeline eliminate the presumably erroneous classification of Cervus? Otherwise, what could explain this inaccuracy in the 12S data?

RESPONSE: Thank you for question. In writing an answer to your question, we’ve done an additional detailed analysis of our data and we’d like to inform you that we’ve made some corrections. 

As reported, we detected an OTU with a sequence highly similar to that of Cervus, which is thought to extinct in the wild in South Korea. However, by further literature survey, we found that they may still exist. In this regard, we have added the following information to the revised manuscript:

“We detected an OTU with a sequence highly similar to that of Cervus (Fig. 2 and Table 1), which is believed to be extinct in the wild in South Korea [6]. The detected sequence is phylogenetically distant from those of Hydropotes, an extant deer species in Korea, so the misidentification of Hydropotes is unlikely (S4 and S5 Figs). We consider that the detected sequence is from Cervus, as they may still exist in South Korea. In fact, species such as Cervus nippon (sika deer), Cervus elaphus (red deer), and Cervus canadensis (elk) are imported from abroad and farmed for their antlers and blood [7, 8]. It has been also reported that the sika deer has been released into the wild by religious events, and about 50 to 100 individuals are known to inhabit Mt. Songni near Chungcheongbuk-do and Gyeongsangbuk-do [6, 9]. There have also been reports of some deer escaping farmland or being illegally released by farmers in Korea [10]. Therefore, albeit in very low numbers, there is a possibility that deer belonging to Cervus inhabit the wild environment of Korea, and it is possible that they were hunted by leopard cats.” Lines 347-359

We also detected the genus Euryoryzomys that is not known to inhabit Korea. We consider that the detection of Euryoryzomys was a misidentification of Myodes. We have added sentences below explaining why. 

“The genus Euryoryzomys detected (Table 1) is not known to inhabit Korea. We consider that the detection of Euryoryzomys was a misidentification of Myodes because these two rodents have very similar vertebrate mitochondrial 12S rRNA gene sequences (S6 Fig). One hypothesis is that the intra-genus variability in the vertebrate mitochondrial 16S gene sequences within the genus Myodes may be comparably large to the inter-genus variability between Myodes and Euryoryzomys. However, this needs to be clarified in future research.” Lines 361-366

9. L321-335: this paragraph would be better placed in the next section (non-mammalian vertebrates & invertebrates)

RESPONSE: Thank you for your suggestion. Following your suggestion, the last paragraph of Discussion 4.1 (lines 325 through 339) has been moved to the first paragraph of Discussion 4.2.

10. L380: please elaborate on the idea that plants were ingested secondarily by eating herbivorous prey. This is an interesting idea, is there any evidence in other studies that metagenomic sequencing of carnivore diets can detect plant DNA ingested by prey? What type of plants do the prey of leopard cats consume, and did you see any of those plant taxa in the metagenomic data?

RESPONSE: To the best of our knowledge, no studies using shotgun metagenomic sequencing to investigate plant feeding by carnivores exist. We believe our research is the first. Meanwhile, the plants that leopard cats feed on or secondarily consume are described in the discussion section (Lines 420-432), citing examples of previous studies [11-14] using methods other than metagenomic sequencing. The abundant detection of Magnoliopsida from feces of leopard cats was reported by one of these studies [13], which is consistent with our finding since we also detected plants belonging to the phylum Magnoliopsida abundantly (Fig. 1C). 

We also scrutinized plants identified from leopard cat feces at the genus level by our metagenomic sequencing (S4 Table). Similar to our vertebrate results, the plants identified at the genus level by shotgun metagenomic sequencing are likely biased toward species whose genomes have already been sequenced. It seems that the inaccurate results were because the genomic database of plants as well as animals is far more incomplete than those of bacteria and fungi. As such, the information that can be obtained from metagenomic sequencing may be limited, so the following sentence has been added as a cautionary note.

“Note that similar to the vertebrate results, the plants identified at the genus level by shotgun metagenomic sequencing are likely biased toward species whose genomes have already been sequenced (S4 Table in S1 File). Therefore, to precisely identify plant taxa that were consumed or secondarily ingested by leopard cats, future research should perform sequencing of plant DNA markers (e.g., ITS) for which a broader range of plant taxa are covered in reference databases.” Lines 432-437

11. L420-423: see earlier comment on BLASTX parameters – would adjusting these parameters reduce the number of misclassifications?

RESPONSE: Thank you for your question, again. As answered in your question #5, there are not enough genomes in the reference database (RefSeq). We thought that adjusting the e-value or percent identity cutoff value will not help reduce misclassification.

12. Fig 1A: Rather than using a minimally-informative pie chart, I think it would be better to show a stacked bar chart with the diet composition of each individual sequenced (similar to Fig 3A)

RESPONSE: Thank you for your suggestion. Fig 1A was changed.

13. Fig 2: If I understand correctly, the x-axis on this figure shows phylogenetic relationships among taxa. Please include how this tree was generated in the methods section, or elaborate what is meant by "euclidean distance" in the figure caption.

RESPONSE: Thank you for your question. The dendrogram represents the similarity of log-transformed relative abundance results, not phylogenetic relationships, between samples (x-axis) or between prey animals (y-axis). This is clarified in the figure legend, as follows: 

“The dendrograms represent the similarity (dissimilarity) of log-transformed relative abundance results between samples (x-axis) or prey animals (y-axis). The dissimilarity is represented by branch lengths based on Euclidean distance.” Lines 248-250

References

1. Wilting A, Brodie J, Cheyne S, Hearn A, Lynam A, Mathai J, et al. Prionailurus planiceps. The IUCN Red List of Threatened Species 2015. 2015:e.T18148A50662095.

2. Mukherjee S, Appel A, Duckworth JW, Sanderson J, Dahal S, Willcox DHA, et al. Prionailurus viverrinus. The IUCN Red List of Threatened Species 2016. 2016:e.T18150A50662615.

3. Mukherjee S, Duckwoth J, Silwa A, Appel A, Kittle A. Prionailurus rubiginosus. The IUCN Red List of Threatened Species 2016. 2016:e.T18149A50662471.

4. Keegan KP, Glass EM, Meyer F. MG-RAST, a Metagenomics Service for Analysis of Microbial Community Structure and Function. In: Martin F, Uroz S, editors. Microbial Environmental Genomics (MEG). New York, NY: Springer New York; 2016. p. 207-33.

5. Kumari P, Tripathi BM, Dong K, Eo KY, Lee W-S, Kimura J, et al. The host-specific resistome in environmental feces of Eurasian otters (Lutra lutra) and leopard cats (Prionailurus bengalensis) revealed by metagenomic sequencing. One Health. 2022;14:100385.

6. Jo YS, Baccus JT, Koprowski JL. Mammals of Korea: a review of their taxonomy, distribution and conservation status. Zootaxa. 2018;4522(1):1–216.

7. Kwak WS, Kim KH, Kim CW, Jeon BT, Lee SM. Deer farming in Korea :On-farm survey in Kyung-kee and Chung-buk provinces. Asian Australas J Anim Sci. 1994;7(3):347–55.

8. Kim SW, Kim KW, Park SB, Kim MJ, Yim DG. Quality characteristics and composition of the Longissimus muscle from entire and castrate elk in Korea. Asian Australas J Anim Sci. 2016;29(5):709–15.

9. Kim G-C, Lee Y-H, Jung D-H, Kim K-Y, Kim Y-h, Han H-s, et al. Home range and behavioral characteristics of released the sika deer (Cervus nippon) by using GPS collar in Songnisan National Park. Korean J Environ Ecol. 2016;30(6):962–9.

10. Banjade M, Park SM, Adhikari P, Han SH, Jeong YH, Lee JW, et al. Molecular evidence reveals the sympatric distribution of Cervus nippon yakushimae and Cervus nippon taiouanus on Jeju Island, South Korea. Animals. 2022;12(8):998.

11. Tatara M. Comparative analyses on food habits of Japanese marten, Siberian weasel and leopard cat in the Tsushima islands, Japan. Ecol Res. 1994;9(1):99–107.

12. Lee O, Lee S, Nam D-H, Lee HY. Food habits of the leopard cat (Prionailurus bengalensis euptilurus) in Korea. Mammal Study. 2014;39(1):43–6.

13. Fatima H, Mahmood T, Hennelly LM, Farooq M, Akrim F, Nadeem MS. Spatial distribution and dietary niche breadth of leopard cats (Prionailurus bengalensis) inhabiting Margalla Hills National Park, Pakistan. Zool Stud. 2021;60:34.

14. Xiong M, Shao X, Long Y, Bu H, Zhang D, Wang D, et al. Molecular analysis of vertebrates and plants in scats of leopard cats (Prionailurus bengalensis) in southwest China. J Mammal. 2016;97(4):1054–64.

---

## [Decision Letter · Decision Letter 1]

3 Jan 2023

PONE-D-22-22486R1Combining vertebrate mitochondrial 12S rRNA gene sequencing and shotgun metagenomic sequencing to investigate the diet of the leopard cat (Prionailurus bengalensis) in KoreaPLOS ONE

Dear Dr. Yamamoto,

Thank you for submitting your manuscript to PLOS ONE. After careful consideration, we feel that it has merit but does not fully meet PLOS ONE’s publication criteria as it currently stands. Therefore, we invite you to submit a revised version of the manuscript that addresses the points raised during the review process.

We look forward to receiving your revised manuscript.

Kind regards,

Michael Schubert

Academic Editor

PLOS ONE

Journal Requirements:

Reviewers' comments:

Reviewer's Responses to Questions

**Comments to the Author**

1. If the authors have adequately addressed your comments raised in a previous round of review and you feel that this manuscript is now acceptable for publication, you may indicate that here to bypass the “Comments to the Author” section, enter your conflict of interest statement in the “Confidential to Editor” section, and submit your "Accept" recommendation.

Reviewer #1: All comments have been addressed

Reviewer #2: (No Response)

2. Is the manuscript technically sound, and do the data support the conclusions?

Reviewer #1: Yes

Reviewer #2: Yes

3. Has the statistical analysis been performed appropriately and rigorously? 

Reviewer #1: Yes

Reviewer #2: Yes

4. Have the authors made all data underlying the findings in their manuscript fully available?

Reviewer #1: Yes

Reviewer #2: Yes

5. Is the manuscript presented in an intelligible fashion and written in standard English?

Reviewer #1: Yes

Reviewer #2: Yes

6. Review Comments to the Author

Reviewer #1: The authors have addressed my previous comments, and the current form of the manuscript is suitable for publication.

Reviewer #2: The authors have addressed the majority of my comments satisfactorily. However, I believe they misunderstood my comment #6 (pasted below):

L244-249: It would be interesting to know whether there was any correlation between the 12S results and metagenomic results within each individual, i.e. using a mantel test or similar multivariate test.

The authors' response:

Thank you for your suggestion. If we understand your suggestion correctly, your suggestion is to examine the leopard cat individual from which each fecal sample was derived, and then compare the results of vertebrate mitochondrial 12S rRNA gene sequencing and shotgun metagenomic sequencing within each leopard cat individual. As answered in your third question, we tried to determine whether the samples used for shotgun metagenomic sequencing were derived from the same individual. However, due to limited recovery and shorter size of the shotgun metagenomic sequencing reads, it is not feasible to assemble these reads into larger genomic contigs to perform such analysis.

My clarification:

Conducting a mantel (or similar) test should not require the metagenomic sequencing results to be assembled into longer contigs. On the contrary, these tests are often used to compare distance matrices composed of different data types. They could be used to determine whether the results from the eleven cats analyzed by both 12S and metagenomic sequencing result in similar euclidean distances. I recommend the authors look at how Li et al. 2017 (doi: 10.3389/fmicb.2017.00454) used mantel tests to determine correlations between different distances including host phylogenetic distance, gut microbiota, and metabolites. A similar approach could be used to determine whether 12S and metagenomic sequencing results were correlated within each individual.

7. PLOS authors have the option to publish the peer review history of their article (what does this mean?). If published, this will include your full peer review and any attached files.

Reviewer #1: No

Reviewer #2: **Yes: **Claire Couch

---

## [Author Response · Author response to Decision Letter 1]

8 Jan 2023

Please find our responses to the reviewers’ comments below. The parts changed are marked in the revised manuscript, and the line numbers in our responses refer to the revised manuscript with track changes.

Reviewer #1:

1. The authors have addressed my previous comments, and the current form of the manuscript is suitable for publication.

RESPONSE: Thank you for your interest in reviewing our submitted paper and the paper we have revised to reflect your opinion. We were able to study a lot while studying your comments and found that the revised paper was of better quality. Once again, thank you for your interest in our paper and for your time.

Reviewer #2:

1. The authors have addressed the majority of my comments satisfactorily. 

RESPONSE: Thank you for your interest in reviewing our submitted paper and the paper we have revised to reflect your opinion. We were able to study a lot while studying your comments and found that the revised paper was of better quality. Once again, thank you for your interest in our paper and for your time.

2. However, I believe they misunderstood my comment #6 (pasted below):

L244-249: It would be interesting to know whether there was any correlation between the 12S results and metagenomic results within each individual, i.e. using a mantel test or similar multivariate test.

The authors' response:

Thank you for your suggestion. If we understand your suggestion correctly, your suggestion is to examine the leopard cat individual from which each fecal sample was derived, and then compare the results of vertebrate mitochondrial 12S rRNA gene sequencing and shotgun metagenomic sequencing within each leopard cat individual. As answered in your third question, we tried to determine whether the samples used for shotgun metagenomic sequencing were derived from the same individual. However, due to limited recovery and shorter size of the shotgun metagenomic sequencing reads, it is not feasible to assemble these reads into larger genomic contigs to perform such analysis.

My clarification:

Conducting a mantel (or similar) test should not require the metagenomic sequencing results to be assembled into longer contigs. On the contrary, these tests are often used to compare distance matrices composed of different data types. They could be used to determine whether the results from the eleven cats analyzed by both 12S and metagenomic sequencing result in similar euclidean distances. I recommend the authors look at how Li et al. 2017 (doi: 10.3389/fmicb.2017.00454) used mantel tests to determine correlations between different distances including host phylogenetic distance, gut microbiota, and metabolites. A similar approach could be used to determine whether 12S and metagenomic sequencing results were correlated within each individual.

RESPONSE: Thank you for your clarification. We are sorry that we misunderstood your question. According to your suggestion, the Mantel test was conducted. 

To proceed with the Mantel test, first, distance matrices were obtained at the class level from 12S and metagenomic sequencing data. These distance matrices, based on Bray–Curtis dissimilarity, are the same ones we used to create Fig 3C. Then, the Mantel test was conducted based on Spearman’s rank correlation analysis, in order to evaluate the correlation between distance matrices of 12S and metagenomic sequencing. As a result of the Mantel test, no correlation was found between the two sequencing data at the class level. This seems to be caused by the incompleteness of the reference database used for metagenomic sequencing, as discussed in our paper. In addition, the performance of the Mantel test and its results have been added to the paper as follows.

“Additionally, the Mantel test was performed based on the Bray–Curtis dissimilarity with 10,000 permutations, in order to evaluate the Spearman’s rank correlation between the distance matrices of shotgun metagenomic sequencing and vertebrate mitochondrial 12S rRNA gene sequencing.” Lines 194-197

“No correlation was found between the distance matrices of shotgun metagenomic sequencing and vertebrate mitochondrial 12S rRNA gene sequencing at the class level (r = −0.05, p > 0.05; Mantel test).” Lines 272-274

---

## [Editor Report · Decision Letter 2]

19 Jan 2023

Combining vertebrate mitochondrial 12S rRNA gene sequencing and shotgun metagenomic sequencing to investigate the diet of the leopard cat (Prionailurus bengalensis) in Korea

PONE-D-22-22486R2

Dear Dr. Yamamoto,

We’re pleased to inform you that your manuscript has been judged scientifically suitable for publication and will be formally accepted for publication once it meets all outstanding technical requirements.

Kind regards,

Michael Schubert

Academic Editor

PLOS ONE

---

## [Editor Report · Acceptance letter]

23 Jan 2023

PONE-D-22-22486R2 

Combining vertebrate mitochondrial 12S rRNA gene sequencing and shotgun metagenomic sequencing to investigate the diet of the leopard cat *(Prionailurus bengalensis)* in Korea 

Dear Dr. Yamamoto:

I'm pleased to inform you that your manuscript has been deemed suitable for publication in PLOS ONE. Congratulations! Your manuscript is now with our production department. 

Kind regards, 

on behalf of

Dr. Michael Schubert 

Academic Editor

PLOS ONE